# Development and Temperature Correction of Piezoelectric Ceramic Sensor for Traffic Weighing-In-Motion

**DOI:** 10.3390/s23094312

**Published:** 2023-04-27

**Authors:** Hailu Yang, Yue Yang, Guanyi Zhao, Yang Guo, Linbing Wang

**Affiliations:** 1National Center for Materials Service Safety, University of Science and Technology Beijing, Beijing 100083, China; yangyue@ustb.edu.cn (Y.Y.); zhaoguanyi@ustb.edu.cn (G.Z.); guoyang@ustb.edu.cn (Y.G.); 2College of Engineering, University of Georgia, Athens, GA 30602, USA

**Keywords:** piezoelectric sensor, temperature compensation, Weigh-In-Motion, multivariate nonlinear fitting, error correction

## Abstract

Weighing-In-Motion (WIM) technology is one of the main tools for pavement management. It can accurately describe the traffic situation on the road and minimize overload problems. WIM sensors are the core elements of the WIM system. The excellent basic performance of WIMs sensor and its ability to maintain a stable output under different temperature environments are critical to the entire process of WIM. In this study, a WIM sensor was developed, which adopted a PZT-5H piezoelectric ceramic and integrated a temperature probe into the sensor. The designed WIM sensor has the advantages of having a small size, simple structure, high sensitivity, and low cost. A sine loading test was designed to test the basic performance of the piezoelectric sensor by using amplitude scanning and frequency scanning. The test results indicated that the piezoelectric sensor exhibits a clear linear relationship between input load and output voltage under constant environmental temperature. The linear correlation coefficient R^2^ of the fitting line is up to 0.999, and the sensitivity is 4.04858 mV/N at a loading frequency of 2 Hz at room temperature. The sensor has good frequency-independent characteristics. However, the temperature has a significant impact on it. Therefore, the output performance of the piezoelectric ceramic sensor is stabilized under different temperature conditions by using a multivariate nonlinear fitting algorithm for temperature compensation. The fitting result R^2^ is 0.9686, the root mean square error (RMSE) is 0.2497, and temperature correction was achieved. This study has significant implications for the application of piezoelectric ceramic sensors in road WIM systems.

## 1. Introduction

The highway system is an important part of national investment and the core of the transportation system. Its security status has a great impact on social and economic development, people’s production, and life. Vehicle overloading is a problem that seriously affects road traffic safety in China. Its harm to the road surface is mainly reflected in reducing the service life of roads and bridges, bringing negative impacts on road safety and service level [1], increasing road maintenance costs, disrupting the operation order of traffic transportation, and affecting the normal use of roads [2]. Vehicle overloading brings many hazards. There is an urgent need to reduce the phenomenon of overload, monitor and evaluate the traffic volume, and ensure the safety and efficiency of vehicle transportation. The WIM system automatically obtains the vehicle load information through the buried load cells on the road without affecting the normal driving of vehicles, so as to achieve the purpose of overload control [3,4,5]. As the core of the WIM system, the sensor plays a vital role in the whole measurement process. At present, in some applied vehicle WIM systems, according to the different sizes and installation methods, the dynamic weighing sensors can be roughly divided into four categories: strain-type weighing sensors, capacitive weighing sensors, optical fiber weighing sensors, and piezoelectric weighing sensors [4,6]. The WIM system with a piezoelectric sensor as the weighing element has the advantages of having a low cost, convenient installation, and high sensitivity. It has therefore become a new idea for the development of dynamic weighing [7].

Piezoelectric sensors use piezoelectric materials as sensing elements. Piezoelectric materials are functional materials that can convert electrical energy and mechanical energy into each other. The conversion of mechanical energy into electrical energy is called the positive piezoelectric effect, while the conversion of electrical energy into mechanical energy is called the inverse piezoelectric effect. Today, piezoelectric sensors provide sustainable solutions for high-performance, low-power electronic equipment required in aerospace, automotive, biomedical equipment, and many other industrial fields [8,9,10]. According to different piezoelectric materials, piezoelectric sensors can be mainly divided into three categories [11]: piezoelectric film sensors, piezoelectric quartz sensors, and piezoelectric ceramic sensors. Piezoelectric film sensors are made of flexible polyvinylidene fluoride (PVDF) film, which is usually packaged as piezoelectric cables for vehicle information monitoring. Its flexible feature makes it easy for it to be affected by road ruts and potholes, and it is difficult to calibrate for high-precision weighing results [12]. Piezoelectric quartz is a natural piezoelectric material, which is one of the most sensitive, expensive, and fragile piezoelectric materials. It is usually used as the sensitive component of the WIM system in the form of an I-shaped structure. Its measurement accuracy largely depends on the flatness and structure of the road surface. Therefore, the installation and regular calibration costs are high, so the piezoelectric quartz dynamic weighing system is rarely considered in remote areas of developing countries [13]. Piezoelectric sensors with piezoelectric ceramic materials as the core have enabled many research achievements in the field of the health monitoring of pavement structure and energy collection [14]. Piezoelectric ceramics have good electrical and mechanical properties. They have high piezoelectric constants, tens of times higher than piezoelectric quartz and PVDF. In addition, they have good structural compatibility. The compressive strength of piezoelectric ceramics is close to that of basalt, which is much higher than that of PVDF. Therefore, piezoelectric ceramics are suitable for road WIM. The accuracy of a WIM system is affected by many factors, such as vehicle speed, road temperature, and wind speed and direction [15]. In recent years, Song et al. [16] prepared a piezoelectric sensor using 3D printing technology, proving its application prospect in the field of vehicle dynamic weighing. Krzysztof Sekuła et al. [17] used piezoelectric measurement technology to monitor the strain development of deformed bodies and finally used these measurements for pavement load identification, laying a solid foundation for further application of civil engineering infrastructure. Xiong et al. [18] developed a P-WIM dynamic weighing system based on piezoelectric ceramic materials. By analyzing the voltage generated by the P-WIM, the axle load of the vehicle can be determined. Compared with the traditional WIM system, the P-WIM system requires almost no maintenance. The capital investment is reduced by 80%, and the manpower and energy required are also greatly reduced. However, poor temperature stability is a bottleneck in the application of piezoelectric ceramic materials. Researchers have tried to improve the stability of materials through various methods to ensure that they can work reliably at different temperature states. Feifei Guo et al. [19] provided a new design strategy for optimizing the performance of piezoelectric ceramics by doping composite materials. Yanan Zhao et al. [20] explored the temperature characteristics of different piezoelectric composites. Nan Liu et al. [21] modified the temperature-dependent piezoelectric constants and capacitance in the displacement mathematical model of PZT materials, obtaining a PZT displacement model under the influence of temperature and verifying the accuracy of the model. Baral et al. [22] studied the effect of temperature change on the admittance signature of piezo sensors and proposed an algorithm for the vertical and horizontal shift in admittance signatures at various frequency and temperature. Dong et al. [23] proposed a temperature compensation methodology based on an NTC thermistor for improving the measurement accuracy of piezoelectric sensing devices. The NTC thermistor paralleled with the constant resistor and compensation circuit are combined with the piezoelectric disk. The results show that the output voltage has an approximate linear relationship with the applied pressure. Sujan et al. [24] proposed a new design and testing method for pressure sensors that utilizes a dual rotation cut piezoelectric gallium silicate (LGS) crystal resonator for temperature compensation. The sensor can simultaneously measure temperature and pressure and can achieve temperature compensation at high temperatures. Jun Lin et al. [25] analyzed the impact of temperature on the sensitivity and amplitude frequency characteristics of sensors, and based on the theoretical formula model of transfer functions and experimental data, established a temperature drift model of sensors and provided a temperature compensation model for sensitivity.

In this study, a high-sensitivity, low-cost and stable sensor based on PZT-5H piezoelectric ceramic was developed for traffic WIM application. The sensor showed a good linear relationship between the output signal and the load and was insensitive to the load frequency within the low-frequency range. A temperature probe was integrated into the sensor to measure the sensor signal characteristics at different temperatures, and multivariate nonlinear fitting method was conducted to compensate for the impact of temperature on sensor data. This research achievement can enable piezoelectric ceramic sensors to continue to be used for traffic WIM even under temperature changes.

## 2. Materials and Methods

The piezoelectric sensor for traffic dynamic weighing consists of three parts: a piezoelectric sensing unit, a charge amplification module, and a data-acquisition module. The piezoelectric sensing unit generates charge under the action of traffic load. The charge is converted into voltage signal by the charge amplification module, and then the voltage signal is converted into digital signal by the data acquisition module. The digital signal can be analyzed to calculate the load to be used for the weighing-in-motion.

### 2.1. Piezoelectric Ceramic Materials

This research was based on previous research about the piezoelectric sensors [26,27]. The selected piezoelectric ceramic material was PZT-5H (Hongsheng Acoustic Electronic Equipment Co., Ltd., Baoding, China). The typical values of the PZT-5H materials parameters provided by the manufacturer are shown in Table 1. The piezoelectric constant was the most important indicator that influenced the sensitivity of the sensor. The piezoelectric constant represents the electromechanical coupling ability of piezoelectric ceramics. The larger the value, the stronger the coupling ability is, which means a higher sensitivity of the sensing. That is, the larger the value of this parameter within a certain range, the better it is for piezoelectric sensors.

PZT-5H is an ideal sensing material because of its relatively high piezoelectric coefficient. However, its piezoelectric coefficient and dielectric constant are easily affected by temperature, so when PZT-5H is used as a sensing material, temperature compensation has to be performed.

To further confirm the temperature stability of PZT-5H piezoelectric materials, the capacitance and piezoelectric constants of PZT-5H piezoelectric ceramic sheets were tested at different temperatures. Five pieces of PZT-5H piezoelectric ceramic sheet with the same size (φ 20 mm × 1.5 mm) were selected and tested at the temperatures of −10 °C, 5 °C, 20 °C, 35 °C, and 50 °C. All samples were incubated at the set temperature for one hour before being tested. After the set temperature was reached, the capacitance and piezoelectric coefficient of each test sample were measured in turn. The capacitive inductance meter UT603 was used to measure the capacitance C, and the quasi-static piezoelectric coefficient measuring instrument ZJ-6A was used to measure the piezoelectric constant d_33_.

Table 2 and Table 3 show measurement results of capacitance values and piezoelectric coefficients of the PZT-5H piezoelectric ceramics at different temperatures. The test results demonstrate that PZT-5H material parameters are affected by temperature. As shown in Figure 1, within the range of −10 °C to 50 °C, capacitance values and piezoelectric coefficients of PZT-5H are generally positively correlated with temperature.

Temperature has a significant effect on the response of piezoelectrics, as it can affect the charge output, frequency response, and voltage sensitivity of the signal [28]. To study the effect of temperature on the fabricated piezoelectric sensors, a temperature sensor was integrated into the sensor. This integration allowed for convenient collection of the actual internal temperature, providing valuable data. PT-100 is a high-precision thermocouple temperature sensor with a measurement accuracy of ±0.5 °C, a probe length of 25 mm, and an acquisition range from −40 °C to 100 °C. The temperature display device adopted a CHB401 intelligent temperature control instrument with a digital display. Compared with the conventional PID control, it has the characteristics of rapid temperature control, fast response, high accuracy, and ability to display the temperature intelligently in real time. The PT-100 temperature sensor has a low-output impedance and is linear. It has an accurate correction device inside. During use with the CHB401 temperature control instrument, the connection and operation are convenient.

### 2.2. Fabrication of Piezoelectric Sensor

Piezoelectric WIM sensors need to be embedded in the road surface layer when used. To withstand traffic loads, it is also necessary to have environmental resistance, such as waterproof performance, compressive wear resistance, corrosion resistance, etc. The piezoelectric sensor designed in this paper had an overall size of 150 mm × 45 mm × 28 mm, and it was a belt-type structure. The packaging material is a glass fiber nylon composite (PA66 + 30% GF), which has high strength and rigidity. The tensile strength is more than 100 MPa and the elastic modulus is over 5900 MPa, and has good bending resistance, compression resistance, and torsion resistance. PA66 + 30% GF is very suitable for making various structural parts. In addition, it also has good heat resistance, chemical corrosion resistance, and fatigue resistance. The filling material is a two-component epoxy resin (k-9741). After hardening, this material has good waterproof and insulation properties and is not easy to crack after curing. It also has good sealing and moisture resistance. The positions of the piezoelectric ceramic chip and the temperature sensor were reserved in the piezoelectric packaging slot. The specific internal layout is shown in Figure 2a, and it is shown in Figure 2b after filling and hardening. The fabrication process of the piezoelectric ceramic sensor was as follows:Firstly, a stainless-steel gasket was placed into the reserved slot, and a piezoelectric ceramic sheet was attached to the gasket with high-performance glue.Secondly, a wire was connected to the positive and negative electrodes of the piezoelectric sheet and fixed in the slot on the slot wall, with the remaining wire extending out of the slot.Thirdly, a temperature sensor was fixed at the reserved position at one end of the piezoelectric tank through threaded connection; fourthly, flexible electronic silica gel was poured around the piezoelectric ceramic along the side wall of the piezoelectric tank.Finally, after the flexible electronic gel solidified, two-component epoxy resin was poured into the slot.

### 2.3. Experimental Scheme

WIM piezoelectric sensors are used for weighing traffic vehicle loads, with traffic parameters mainly including vehicle speed and axle load, corresponding to loading frequency and load amplitude. Environmental parameters include temperature, rainfall, wind speed, etc. This study focuses on analyzing the impact of temperature. This study conducted testing and analysis through indoor experiments, with experimental parameters including load size, frequency, and temperature.

#### 2.3.1. Test Condition

The loading equipment for piezoelectric sensor performance test was an electro-hydraulic servo fatigue testing machine with a temperature-control box from the test equipment company Walter + Bai in Switzerland. The maximum load is ±100 kN, the load frequency can reach 20 Hz, and the range of the temperature is from −150 °C to 350 °C. Stainless steel strip gaskets were added between the upper and lower parts of the test piece to ensure uniformity. The loading structure is shown in Figure 3.

The test applied different loading conditions through the electronic hydraulic servo machine and then transmitted the load to the piezoelectric ceramic sensor. Under the dynamic load, the piezoelectric sensor exerted force on the piezoelectric sensing element and generated a voltage signal, which was acquired through the data acquisition system. The data acquisition process is shown in Figure 4.

As shown in Figure 5, the data acquisition device consisted of an oscilloscope and a charge amplifier. A digital meter CHB401 was used as the temperature display device during the test. The digital oscilloscope type was DPO2024 from Tektronix with the main parameters as follows:Number of Channels: 4;Bandwidth: 200 MHz;Sampling Rate:1 GSa/s;Record Length: 1 M points.

The charge amplifier adopted an LZ1105-16 with the main parameters as follows:Measuring charge ranges: ±10^7^ pC;Output voltage ranges: ±10,000 mV;Gain factor: 0.1, 0.2, 0.5, 1, 2, 5, 10;Sensitivity: 100 pC/mV;Measurement uncertainty: 0.5%;Frequency range: 0.1–200,000 Hz.

The actual waveform information collected by the oscilloscope is shown in Figure 6. The charge amplifier converted the charge signal generated by the piezoelectric ceramic sensor into a voltage signal and amplified it. Moreover, it prevented the attenuation and loss of the piezoelectric output signal. In this experiment, the charge amplifier adjusted the gain to one time, and the corresponding output was 100 pC/mV.

#### 2.3.2. Design of Experimental Parameters

(1)Temperature

Beijing has a temperate continental climate, with the annual maximum temperature ranging from 35 °C to 40 °C and the annual minimum temperature generally above −10 °C. To investigate the effect of temperature on the output signals of piezoelectric sensors, this test was designed with five test temperature intervals of 15 °C within the range of −10 °C to 50 °C. Specifically, it was divided into −10 °C, 5 °C, 20 °C, 35 °C, and 50 °C. Each temperature lasts for 1 h, and the data are recorded after the set temperature is reached.

(2)Load

According to Specifications for Design of Highway Asphalt Pavement (JTG D50-2017), the maximum pressure of vehicles on highway pavement should not exceed 0.7 MPa, which corresponds to a force of 4.725 kN applied on the surface of the piezoelectric sensor. However, to better reflect the impact of vehicles on the road in the traffic field, the input load range should be expanded when studying the relationship between the output voltage of the piezoelectric sensor and the input load. The input load range in this test was 5–25 kN, with an interval of 2 kN and a total of 11 levels. The width of the piezoelectric sensor designed in this paper was 0.045 m, which was about one-fifth of the width of a typical tire–road contact model. Therefore, the test load could represent a wheel load of 25–125 kN. In a previous study, Yang [29] studied the fatigue performance of piezoelectric materials under high loads. Without packaging materials, a load amplitude of 7 kN was directly applied to the piezoelectric material PZT-5H for 100,000 cycles, and the output voltage did not decrease. Based on the cross-sectional area ratio of piezoelectric ceramic to packaging structures, the maximum load acting on the ceramic was approximately 1.16 kN, which is much lower than the previously studied one. Therefore, the piezoelectric sensor developed in this study would have good fatigue performance under traffic loads.

(3)Frequency

The loading frequency was related to the vehicle driving speed. The loading frequency of 10 Hz was approximately equivalent to the driving speed of 60–65 km/h [30]. Considering the stability of the loading equipment, the loading frequency in this test ranged from 2 Hz to 10 Hz with an interval of 2 Hz. The specific loading frequencies were 2 Hz, 4 Hz, 6 Hz, 8 Hz, and 10 Hz, corresponding to the actual speeds of 10–65 km/h.

#### 2.3.3. Loading Scheme

The performance of the piezoelectric sensor is evaluated using a sine loading scheme with two loading modes: amplitude scanning and frequency scanning. In the amplitude scanning mode, the load amplitude varies from 5 to 25 kN while the loading frequency is fixed. In the frequency scanning mode, the loading frequency varies from 2 to 10 Hz while the load amplitude is fixed. In this study, the frequency-scanning mode was adopted. The tests were conducted at different temperature values in sequence. The testing process was as follows:

Step 1: Install the fabricated piezoelectric sensor sample on the testing machine, adjust the loading position, and then close the door of the temperature-control box;

Step 2: Set the temperature to a start temperature of −10 °C and hold for it for one h;

Step 3: Adopt frequency scanning mode, and set the load amplitude to 5 kN;

Step 4: Set the loading frequency to 2, 4, 6, 8, and 10 Hz in sequence, continuously load each frequency for 1 min, and record the voltage data; 

Step 5: Set the load amplitude to increase by 2 kN, repeat steps 3 and 4 until the load amplitude reaches 25 kN; 

Step 6: Set the temperature to increase by 15 °C and hold for one hour. Then, repeat steps 3–5. Until the loading is completed at a temperature of 50 °C;

Finally, at the end of the experiment, stop loading and cool the test box to room temperature.

## 3. Results and Analysis

### 3.1. Basic Performance Analysis

#### 3.1.1. Linearity and Sensitivity Analysis

Linearity is a crucial performance parameter of piezoelectric sensors and a fundamental requirement for their practical application. It indicates the degree of deviation of the actual curve between the input load and the output voltage of the sensor from the fitted straight line. The higher the linear correlation coefficient, the better the performance of the sensor. In traffic dynamic monitoring, linearity is an essential factor for ensuring the accuracy of vehicle dynamic weighing.

The sensitivity of the piezoelectric sensor is determined by the slope of the linear fitting equation between the input load and the output voltage. The higher the slope value of the fitted linear equation, the better the piezoelectric response characteristics of the sensor, resulting in higher sensitivity and larger output voltage signal.

The test design employed an amplitude scanning loading system to examine the linearity and sensitivity of the piezoelectric sensor under identical loading conditions. The loading frequency was set at 2 Hz, 4 Hz, 6 Hz, 8 Hz, or 10 Hz, and the load amplitude increased gradually from 5 kN to 25 kN in increments of 2 kN. The linearity of the sensor was evaluated by using the linear correlation coefficient of its output signal fitting line. Figure 7 illustrates the linear fitting relationship between the output voltage and input load of piezoelectric sensor at different loading frequencies at room temperature (20 °C). It can be observed that the piezoelectric sensor exhibits excellent piezoelectric output performance. There is a clear linear correlation between input load and output voltage at various loading frequencies. The linear correlation coefficient of fitting line exceeds 0.999, and the sensitivity is 4.04858 mV/N (at 2 Hz). Under the same charge amplification factor, the sensitivity of the sensor developed in this study is 100 times higher than that of the quartz sensor and dozens of times higher than that of the PVDF thin film sensor.

#### 3.1.2. Frequency Independent Characteristic Analysis

Frequency independence is another key characteristic of sensor dynamic response, which reflects the stability of piezoelectric sensor output performance when the input frequency changes under a constant load. When the encapsulated piezoelectric sensor is embedded in the pavement structure, it will be subjected to the same traffic loading frequency as the pavement engineering structure. The more stable the voltage output of the piezoelectric sensor is under different dynamic load frequencies, the better its frequency independence and working performance are.

The test design uses a frequency scanning method to analyze the frequency-independent characteristics of piezoelectric sensors under different temperature conditions. Figure 8 depicts the relationship between the output voltage and frequency under various load amplitudes at room temperature. It can be seen that, in the frequency range of 2–10 Hz, as the load increases, the output voltage of the sensor remains essentially unchanged when load frequency varies, demonstrating good frequency independence. In other words, load frequency does not affect output results of the piezoelectric ceramic sensor. Therefore, the developed sensor is suitable for different vehicle speeds in the traffic weighing process.

#### 3.1.3. Temperature Stability Analysis

In road traffic, the road surface temperature varies significantly due to the alternation of day and night and seasonal changes. Therefore, the temperature stability of the piezoelectric sensing unit must be considered in applications. Piezoelectric ceramics are temperature-sensitive materials, and the piezoelectric coefficient is highly affected by temperature. The load distribution characteristics of the entire piezoelectric sensing unit after packaging are complex, so its stress state is also influenced by the ambient temperature.

Figure 9 shows that the load and output voltage trends were quite different under different temperatures, indicating that the effect of temperature on the piezoelectric sensing unit cannot be ignored. The voltage and load are positively correlated at each temperature but not absolutely linearly correlated. When the temperature is within the range of −10 to 20 °C, the slope of the fitting curve gradually increases as the temperature increases. However, from 20 °C to 50 °C, the slope actually decreases. Temperature has a significant impact on piezoelectric ceramic sensors and requires temperature correction. At any temperature, the output signal of the piezoelectric sensor is close to a linear positive correlation with the load. However, the correlation coefficient varies with temperature.

### 3.2. Temperature Compensation Algorithm

The output results of the piezoelectric sensor can be approximated as a linear process in the temperature range of −10 °C to 50 °C. To mitigate the negative effects of temperature, this paper proposes a mathematical model of the piezoelectric ceramic sensor temperature characteristics and compensates the piezoelectric ceramic sensor with a multivariate nonlinear fitting algorithm. The mathematical model of temperature compensation is given by Equation (1).
(1)$$U=\left(k_{Tem}T+k_{0}\right)F$$
where $k_{Tem}$ is the sensitivity temperature coefficient of the piezoelectric sensor; T is the measurement temperature; $k_{0}^{}$ is the sensitivity of the piezoelectric sensor at 0 °C; and F is the force applied on the sensor.

Because the frequency effect is relatively small, according to Equation (1), this paper fit a curved surface to 55 groups of output voltage data of the piezoelectric sensor at different temperatures, with the loading frequency at 2 Hz and the load varying from 5 kN to 25 kN. The fitting results are presented in Figure 10. During the fitting process, *k_T__em_*= 0.000475563 and *k*_0_ = 0.1905. The fitting equation is given by Equation (2).
(2)U=(0.000475563T+0.1905)F

The multivariate nonlinear fitting algorithm used in this study can effectively correct the temperature error of piezoelectric ceramic sensors. The correlation coefficient R^2^ between the fitting formula and experimental data is 0.96798. As shown in Table 4, in most cases, the relative error between the fitted data and the measured data is less than 15%. At lower force conditions (5 kN, 7 kN), the relative error is relatively large.

## 4. Discussion

The PZT-5H piezoelectric ceramic adopted in this study has excellent electromechanical coupling characteristics and is particularly suitable for use in traffic WIM sensors. However, due to the brittle and fragile characteristics of piezoelectric ceramic sheets, they cannot be directly placed in road materials and structures. Through encapsulation design, its applicability has been improved so that it can be applied to road traffic Weigh-In-Motion sensors. In addition to having high sensitivity, PZT-5H also has strong temperature sensitivity. By pouring and encapsulating temperature sensors and piezoelectric ceramics together, the temperature of piezoelectric ceramics can be obtained under any working condition. In this paper, multivariate nonlinear fitting was used for rapid temperature correction. Sensor parameter calibration is also easy.

In previous studies, WIM systems also included temperature data collection, mostly using small weather stations to obtain air temperature or burying temperature sensors in the pavement to obtain the temperature inside the pavement structure. In this study, a temperature sensor was packaged with a piezoelectric sensing unit. Compared with other solutions, it can better reflect the temperature state of the sensing unit, so the sensing data can be better corrected. In this study, the simplest linear fitting method was adopted, and a very good correction effect was obtained, which was very convenient for engineering applications. 

However, insufficient consideration was given to the temperature characteristics of the encapsulation material in this study. Moreover, since the piezoelectric weighing sensor is to be embedded in the road surface, there exists a deviation between environmental temperature and sensor temperature. Changes in road material and encapsulation material temperatures will affect the stress state of ceramic sheets and thus affect measurement accuracy. This is a complex nonlinear system that can be calibrated using machine learning methods. The temperature range (−10 °C to 50 °C) set in this study mainly focused on Beijing’s climate characteristics. The applicability to a wider temperature range needs further research.

## 5. Conclusions

The basic performance of the WIM sensor should be investigated before its practical application in the project. It should have good environmental adaptability, be able to cope with complex actual working conditions, and have stable output performance. The main conclusions are as follows:(1)The piezoelectric ceramic sensor designed in this paper has a built-in temperature sensor, which offers the advantages of high sensitivity, simple structure, small size, and ability to detect the ambient temperature of the road surface.(2)The indoor sinusoidal loading test was conducted to examine the basic performance of the piezoelectric ceramic sensor. The test results indicate that there is a clear linear relationship between the input load and output voltage of the sensing system at a room temperature. The linear correlation coefficient of the fitted straight line exceeds 0.999, and the sensitivity is 4.04858 mV/N. It exhibits excellent piezoelectric output performance and can be used as a WIM sensor.(3)To address the temperature drift phenomenon of the piezoelectric ceramic sensor, a multivariate nonlinear fitting method was employed for temperature compensation. The fitting result R^2^ is 0.9686, which achieves a good compensation effect and, in most working conditions, a relative error within 15%.

## Figures and Tables

**Figure 1 sensors-23-04312-f001:**
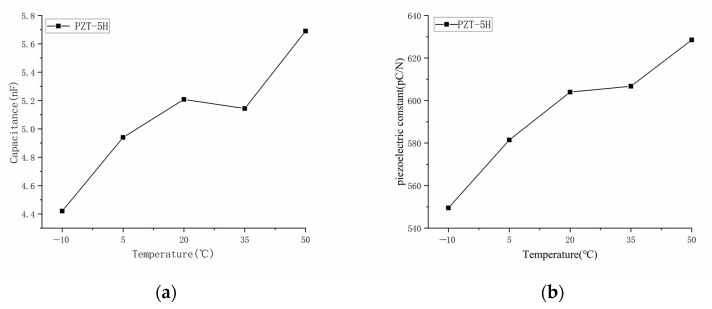
Temperature characteristics of PZT-5H. (**a**) Capacitance; (**b**) piezoelectric constant.

**Figure 2 sensors-23-04312-f002:**
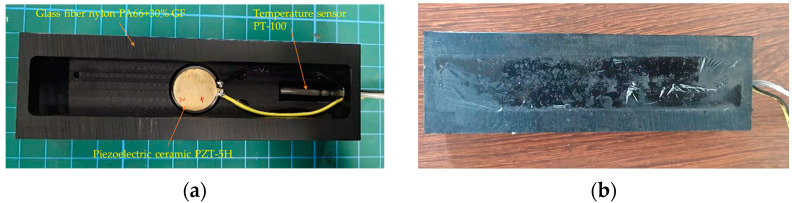
The WIM sensor. (**a**) Internal structure; (**b**) fabricated sensor.

**Figure 3 sensors-23-04312-f003:**
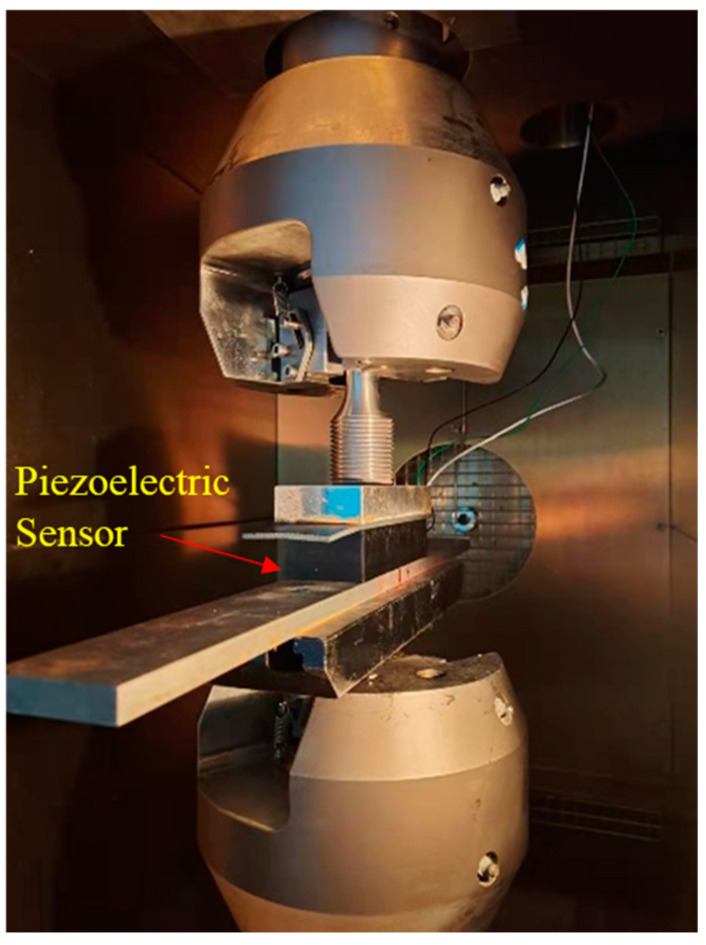
Loading structure.

**Figure 4 sensors-23-04312-f004:**
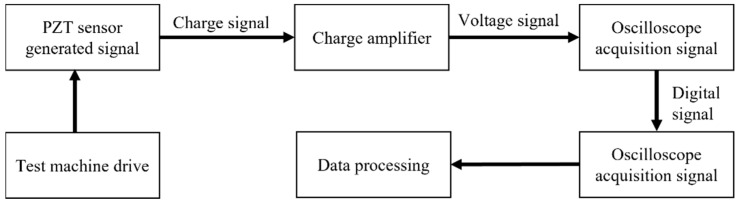
Data acquisition flow.

**Figure 5 sensors-23-04312-f005:**
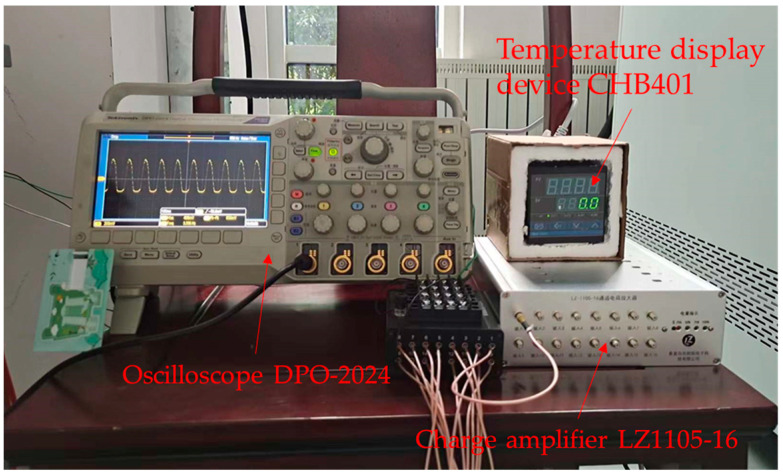
Data acquisition device.

**Figure 6 sensors-23-04312-f006:**
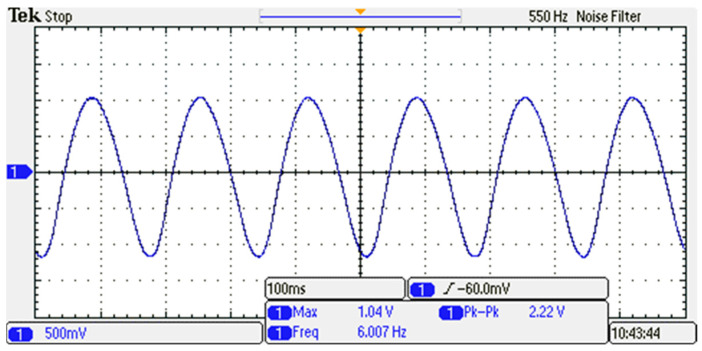
Oscilloscope Acquires Waveform Information.

**Figure 7 sensors-23-04312-f007:**
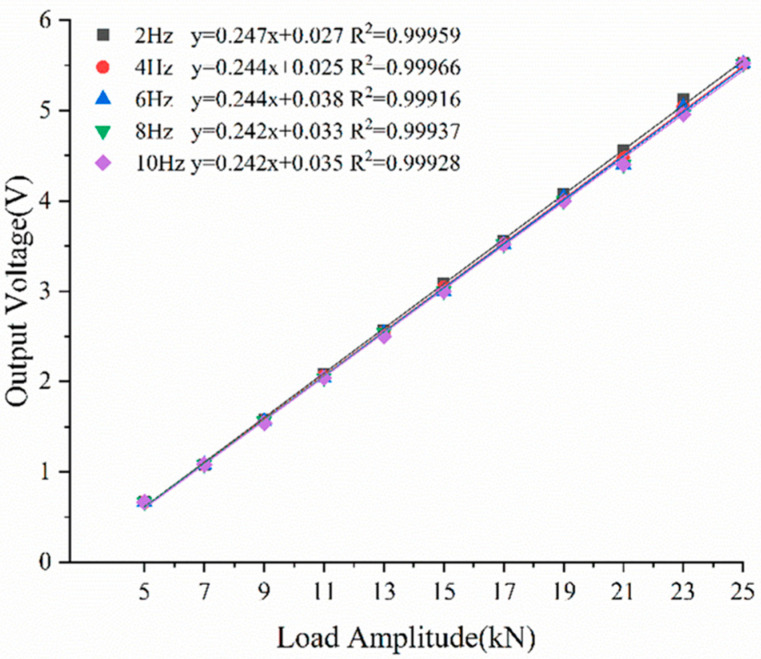
Relation between Output Voltage and Load Amplitude under Different Loading Frequencies.

**Figure 8 sensors-23-04312-f008:**
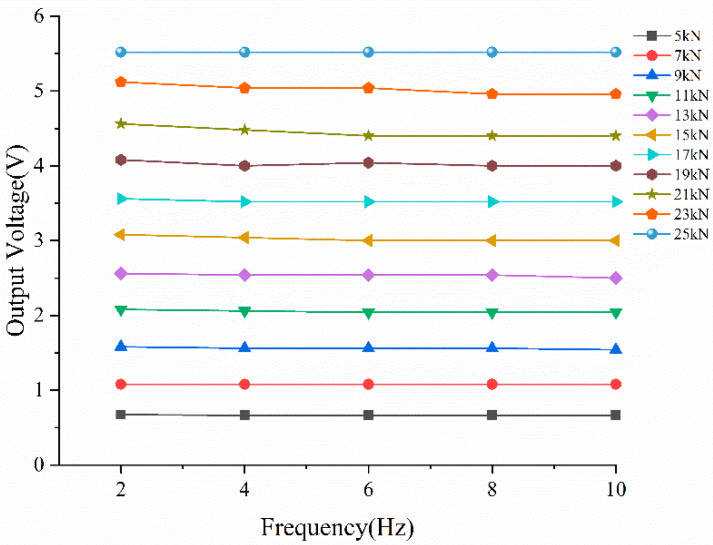
Frequency-independent characteristic analysis of piezoelectric ceramic sensor.

**Figure 9 sensors-23-04312-f009:**
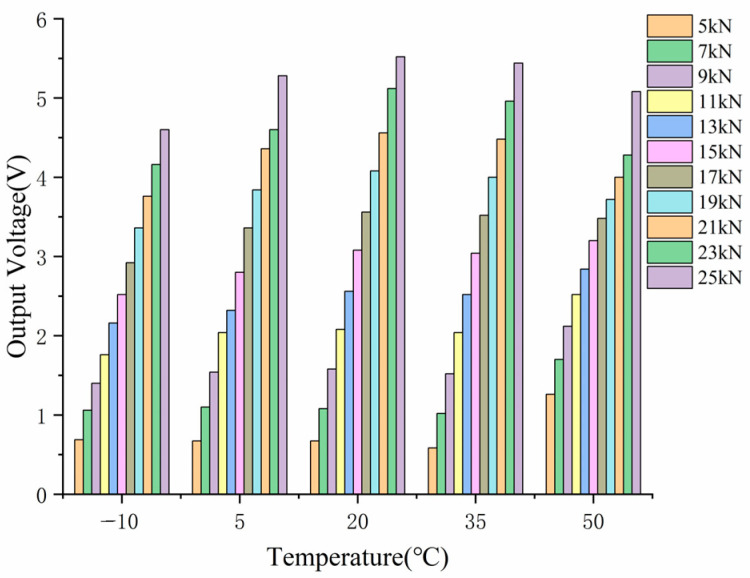
Output Results of Piezoelectric Sensor at Different Temperatures before Compensation.

**Figure 10 sensors-23-04312-f010:**
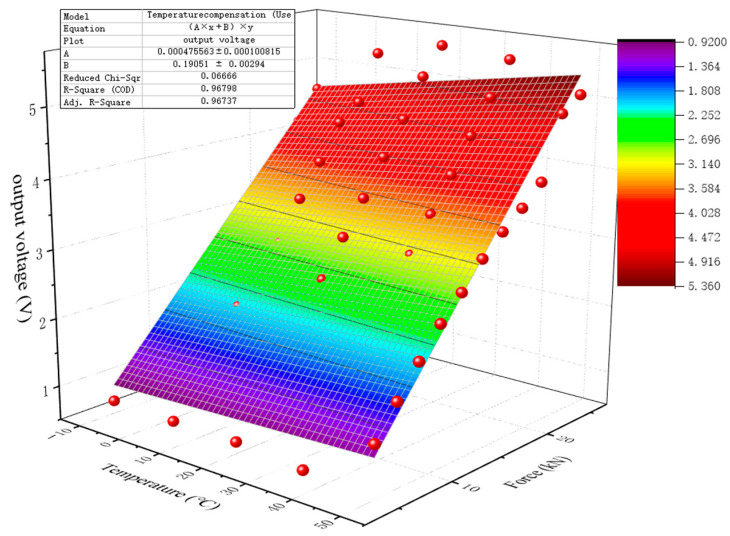
Fitting Results of the Multivariate Nonlinear Fitting Algorithm.

**Table 1 sensors-23-04312-t001:** Typical values of piezoelectric material parameters.

Parameters	Symbols	Values	Units
Piezoelectric constant	$d_{33}$	415	pC/N
Relative dielectric constant	$\varepsilon_{33}^{T}$	2100	-
Curie temperature	$T_{c}$	260	°C
Density	ρ	7.5	10^3^ kg/m^3^
Elastic modulus	E	117	GPa
Electromechanical coupling coefficient	*k* _T_	>0.7	-

**Table 2 sensors-23-04312-t002:** Test results of piezoelectric ceramic capacitance (Unit: nF).

Number	−10 °C	5 °C	20 °C	35 °C	50 °C
1	4.44	4.91	5.31	5.17	5.68
2	4.41	5.04	5.30	5.22	5.72
3	4.44	4.84	5.10	5.03	5.61
4	4.37	5.00	5.02	5.13	5.76
5	4.44	4.91	5.31	5.17	5.68
Average	4.42	4.94	5.208	5.144	5.69

**Table 3 sensors-23-04312-t003:** Test Results of Piezoelectric Constants at Different Temperatures (Unit: pc/N).

Number	−10 °C	5 °C	20 °C	35 °C	50 °C
1	560.1	593.6	622.3	615.2	622.2
2	551.8	598.5	602.9	607.4	633.5
3	542.4	555.7	584.3	591.6	614.8
4	534.5	570.5	606.8	598.2	627.4
5	558.7	589.0	603.6	621.2	644.6
Average	549.5	581.46	603.98	606.72	628.5

**Table 4 sensors-23-04312-t004:** Comparison Between Fitting Results and Measurement Results.

Force	Measured Output Voltage (V)					Fitted Output Voltage (V)					Relative Error				
(kN)	−10 °C	5 °C	20 °C	35 °C	50 °C	−10 °C	5 °C	20 °C	35 °C	50 °C	−10 °C	5 °C	20 °C	35 °C	50 °C
5	0.688	0.672	0.672	0.584	1.26	0.93	0.96	1.00	1.04	1.07	37.32%	43.52%	61.92%	38.39%	11.32%
7	1.06	1.10	1.08	1.02	1.70	1.30	1.35	1.40	1.45	1.50	18.89%	24.56%	35.20%	24.50%	11.75%
9	1.40	1.54	1.58	1.52	2.12	1.67	1.74	1.80	1.86	1.93	9.41%	10.13%	17.73%	16.82%	12.11%
11	1.76	2.04	2.08	2.04	2.52	2.04	2.12	2.20	2.28	2.36	0.19%	2.05%	7.70%	11.83%	12.46%
13	2.16	2.32	2.56	2.52	2.84	2.41	2.51	2.60	2.69	2.79	4.39%	2.26%	3.14%	5.83%	13.06%
15	2.52	2.8	3.08	3.04	3.2	2.79	2.89	3.00	3.11	3.21	0.54%	6.67%	1.29%	3.05%	13.38%
17	2.92	3.36	3.56	3.52	3.48	3.16	3.28	3.40	3.52	3.64	6.92%	8.36%	3.36%	1.18%	13.94%
19	3.36	3.84	4.08	4.00	3.72	3.53	3.66	3.80	3.94	4.07	9.25%	10.81%	4.89%	5.40%	14.57%
21	3.76	4.36	4.56	4.48	4.00	3.90	4.05	4.20	4.35	4.50	12.21%	11.68%	6.13%	7.82%	14.98%
23	4.16	4.6	5.12	4.96	4.88	4.27	4.44	4.60	4.76	4.93	7.88%	14.86%	7.02%	2.33%	13.45%
25	4.6	5.28	5.52	5.44	5.08	4.64	4.82	5.00	5.18	5.36	13.83%	13.22%	7.96%	1.82%	14.04%

## Data Availability

All datasets used in this study are discussed in Section 3. They are publicly available and cited in the list of references.

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
