# Peer review of "Development and Temperature Correction of Piezoelectric Ceramic Sensor for Traffic Weighing-In-Motion"

_sensors, 2023, doi:10.3390/s23094312_

Round 1

Reviewer 1 Report

the comments and suggestions for authors are in the unloaded file.

Author Response

Thanks very much for your review comments. The authors have carefully read the review comments, revised and reponsed one by one.

Reviewer 2 Report

This paper investigates a WIM sensor using PZT-5H piezoelectric ceramic with an integrated temperature probe. The manuscript can be revised based on the following comments:

1. Are the values in Table 1 (e.g., d33, Qm) measured using the PZT-5H of the fabricated prototype?

2. The piezoelectric constant and the piezoelectric ceramic capacitance vary with the temperature based on Table 2 and Table 3. It is suggested using figures to reflect the relationships.

3. The section of conclusion should include the innovation of the work and its differences compared with the previous study.

4. Please provide a figure to reflect the relationship between output voltage and load amplitude under different temperatures.

Author Response

(The authors gave the same response as above.)

Reviewer 3 Report

The authors successfully developed a piezoelectric ceramic sensor for traffic weighing-in-motion (WIM) application, and systematically examined its performance under different loading frequencies, loading amplitudes, and temperatures. The experiment is well conducted, and the manuscript is well organized. However, there are several issues as listed below that need to be addressed before considering publication in this journal.

1.     In table 2 and 3, the capacitance and piezoelectric constants seem not to be positively correlated with temperature in the range of 20-35 °C. It would be better to describe it in detail.

2.     In line 272, the sensitivity obtained is 0.247 KN/V (at 2Hz). How sensitive is it compared with other piezoelectric materials?

3.     In figure 8, the load and output voltage show different trends under different temperatures. It would be better to describe those trends.

4.     The polynomial regression algorithm in this work compensates the temperature drifts using 275 groups of fitting data. This result should be further validated with testing data.

5.     In line 16, rephrase the sentence “A kind of… cost.”

6.     In line 37 and 38, “increasing road maintenance costs” and “leading to the increase of road maintenance costs” are redundant.

7.     In line 49, “sensors [4, 6], ” should be “sensors [4, 6].”

8.     In line 101 and 102, rephrase “Make it still … conditions.”

9.     In line 129, “20 °C. 35 °C” should be “20 °C, 35 °C”

Author Response

(The authors gave the same response as above.)

Reviewer 4 Report

The study presented in the manuscript aimed to design and test a piezoelectric sensor with temperature compensation for Weigh-In-Motion (WIM) applications. The background and motivation of the work are described quite well, and the results are presented in a well-organized and logical manner. Basically, the authors investigated the performance of the sensor and addressed the temperature drift phenomenon through a polynomial fitting for temperature compensation. The results show that the piezoelectric ceramic sensor has a very good piezoelectric output performance and can be used as a dynamic weighing sensor, and its output performance can be stabilized under different temperature conditions. However, there are some aspects of the manuscript that need to be addressed. 

1. The novelty of the study is not clearly highlighted. The authors should emphasize the contribution of the study and provide a comparison with other works, specifically with previously published works by the authors. 

2. The manuscript would benefit from more explaining figures clearly illustrating the experimental setup and results. 

3. The literature review needs to be enriched by adding more relevant research related to piezoelectric sensors and their temperature compensation techniques for WIM applications.

4. Have the authors investigated the long-term stability of the sensor output under different temperature conditions?

5. How does the encapsulation design ensure the applicability of the piezoelectric ceramic sensor to road traffic WIM sensors?

7. Section 2.2: The authors may provide specific details about the instruments used in the experiments, such as the loading equipment and capacitance measurement device.

8. Section 2.2.2: It would be helpful to provide some visual aids to explain the loading protocol to improve the manuscript's readability.

9. Line 322: The sentence needs to be rewritten. 

Overall, the manuscript has the potential to make a valuable contribution to the field of WIM sensors. However, the authors need to address the above-mentioned concerns before it can be accepted for publication.

Author Response

(The authors gave the same response as above.)

Round 2

Reviewer 1 Report

The authors have addressed all my comments and I recommend to accept this work in the present format.

Reviewer 2 Report

The paper can be published.

Reviewer 3 Report

The authors well addressed all the issues in the revised version. I suggest to publish the manuscript as is.